# After-Death Communication: Issues of Nondisclosure and Implications for Treatment

**Kathleen C. Pait** [1,*], **Julie J. Exline** [1], **Kenneth I. Pargament** [2] **and Peri Zarrella** [3]

1    Department of Psychological Sciences, Case Western Reserve University, Cleveland, OH 44106-7123, USA; julie.exline@case.edu
2    Department of Psychology, Bowling Green State University, Bowling Green, OH 43403-0001, USA; kpargam@bgsu.edu
3    Independent Scholar, Cold Spring, NY 10516, USA
*    Correspondence: kathleen.pait@case.edu

**Abstract:** After-death communication (ADC) is the phenomenon of perceiving spontaneous and direct contact by a deceased loved one. Evidence suggests that ADC is a widespread human experience, particularly for bereaved individuals. Many people who have ADCs report them to be comforting, suggesting therapeutic potential. However, many individuals in Western cultures choose not to disclose their ADCs to mental health providers, citing fears of pathologization, disenfranchisement due to gender expectations, negative ADC encounters, or minimization by clinicians. For others, ADCs are deeply personal and people may keep the experiences to themselves for fear that providers might explain the ADCs away by framing them in purely psychological terms. As such, there is a paucity of literature on how therapists should best approach the topic of ADC with clients. The following narrative review offers clinical interview and assessment strategies from theoretical papers and empirical studies to guide this process. Clinicians are encouraged to self-reflect on their natural inclinations around ADC, assess general psychological functioning, normalize and validate the ADC experience, assess client feelings and explanations, and inquire about spiritual beliefs. Clinicians can also help clients to explore the meaning and personal significance of their ADCs as well as clients' perceived relationships with the deceased.

**Keywords:** after-death communication; bereavement; grief counseling; afterlife beliefs; spirituality

## 1. Introduction

We will use the term *after-death communication* (ADC) to refer to the phenomenon of perceiving spontaneous and direct contact by a deceased loved one (Guggenheim and Guggenheim 1997). Importantly, this paper does not address questions about the ontological reality or origin of ADC since such questions are beyond the scope of psychology. Importantly, though, ADCs are often perceived as real by experiencers; thus, the approach of this psychological paper is to focus on perceptions of and attributions/beliefs regarding contact with a deceased person. For readers interested in a more detailed review of some of the theoretical perspectives on the origin of ADC, please see Kamp et al. (2020) and Castelnovo et al. (2015).

It has been estimated that between 30–35% of the general population is likely to experience one ADC over the course of their lives (Streit-Horn 2011), though this may be an underestimate due to underreporting (Castelnovo et al. 2015), which this paper later addresses in more detail. Some individuals report more frequent encounters (Elsaesser et al. 2020; Penberthy et al. 2021; Troyer 2014). ADCs appear to be widespread experiences occurring across cultures (see Castelnovo et al. 2015 for review; Jahn and Spencer-Thomas 2014; Sabucedo et al. 2020; Yamamoto et al. 1969), ethnicities (Kalish and Reynolds 1973), ages (Houck 2005), educational levels (Haraldsson et al. 1976; Klugman 2006), and religious orientations (Barbato et al. 1999; Daggett 2005). Literature on the subject suggests that

ADCs occur through a variety of channels (Elsaesser et al. 2020; Penberthy et al. 2021; Woollacott et al. 2021), such as the felt presence of a deceased loved one (Bennett and Bennett 2000; Datson and Marwit 1997; Guggenheim and Guggenheim 1997; Hayes and Leudar 2016; LaGrand 1997; Woollacott et al. 2021); direct sensory experiences including visual, auditory, tactile, or olfactory encounters (Haraldsson et al. 1976; Nowatzki and Kalischuk 2009; Penberthy et al. 2021); symbolic interpretations of natural phenomena (Guggenheim and Guggenheim 1997; LaGrand 1997; Parker 2005); and messages conveyed in dreams (Kalish and Reynolds 1973; Kwilecki 2011; Woollacott et al. 2021).

### 1.1. After-Death Communication and Supernatural Attributions

There are many reasons why some individuals may come to believe that they have been contacted by the departed. Exline and Pait (2022) have proposed that supernatural attributions for events, including ADCs, may be contingent upon several conditions, including the *accessibility*, *plausibility*, and *motivation* to interpret such events as supernatural. (See Exline and Wilt 2023, for a review of supernatural attributions literature using this framing). For example, supernatural explanations may become *accessible* when individuals have experienced attention-grabbing or unusual events, major life stressors, or environmental cues. Supernatural explanations may be *plausible* for some individuals due to preexisting spiritual beliefs, a lack of satisfying natural explanations, or traits such as openness (Kamp et al. 2019), experiential processing (Wilt et al. 2022), schizotypy (Barnes and Gibson 2013), or prolonged grief or loneliness (Kamp et al. 2020). Individuals could also be *motivated* to interpret events as supernatural when they have the desire to make meaning from them or if they have emotional needs, such as relational closure, that could be met with supernatural explanations.

It is easy to apply Exline and Wilt's (2023) conceptual understanding of supernatural attributions to experiences of ADC. The stress of losing a loved one may lead some to consider the possibility of postmortem contact and they might be vigilant for potential signs of such contact (*accessibility*). A prior belief in God, angels, spirits, or ghosts, combined with a belief that these beings actually do communicate with people in the world, may make ADC a *plausible* occurrence (Exline et al. 2021). Finally, the need for relational closure could *motivate* individuals to see ADCs as true messages from deceased loved ones. Relational factors between the mourner and the deceased could also be relevant to motivation. For example, Rees (1971) found ADCs to be more common among widows who reported happy marriages and who had become parents before their spouses had died.

### 1.2. Purpose of Current Review

Despite the variety and prevalence of ADC reports, as well as the common conditions that may lead some to make supernatural attributions for their ADCs, many individuals in Western cultures withhold reports of such experiences from their mental health providers. In a study comprised of a French and English sample, for example, 90% of participants (*n* = 79) reported sharing their ADC experiences only with selected and trusted others, such as friends (Evrard et al. 2021). Some may be concerned that their ADCs will be treated as symptoms of serious mental illness (Sabucedo et al. 2020; Sanger 2009; Troyer 2014) rather than actual supernatural encounters or normal psychological events (see Exline 2021 for a discussion). Consequently, individuals can experience intense feelings of anxiety when they are not able to talk openly about their ADCs or to process the significance of the events (LaGrand 1997).

As a result, there is a paucity of literature on how therapists should best approach and discuss the subject of ADC when clients do bring it up. Kamp et al. (2020) have provided general clinical recommendations for working with ADC in therapeutic settings, such as approaching the topic with nonjudgmental exploration and cultural sensitivity. In addition, the edited volume by Kramer et al. (2012) includes useful resources on how clinicians can approach broader reports of paranormal experiences from clients if and when they arise in treatment settings. However, organizations like the International

Association for Near-Death Studies (IANDS) that provide resources on near-death-related subjects have appealed to the mental health community to learn more about ADC and related phenomena (see https://iands.org/images/stories/pdf_downloads/mdletter.pdf, accessed on 10 January 2023). Thus, the aims of this narrative review are to expand upon the existing, yet limited, clinical literature by providing a more detailed discussion on the roots of ADC nondisclosure and to consolidate information from theoretical papers and empirical studies to provide specific examples of interview and assessment strategies around the topic. In doing so, we hope to provide a framework for clinicians to help clients process and derive the most meaning, comfort, connectedness, and closure from their encounters. Earlier work has found that disclosure of similar exceptional human experiences is significantly and positively associated with increased well-being and decreased illness (see Palmer and Braud 2002).

## 2. After-Death Communication Nondisclosure

In Western cultures, many clients may share a sense of hesitation to divulge their ADCs (Pait and Exline 2023). One of the most cited reasons for client nondisclosure of ADC is the fear of negative reception (Bennett and Bennett 2000; Keen et al. 2013; Nowatzki and Kalischuk 2009; Rees 1971). For example, when examining accounts of ADCs given by widows, Bennett and Bennett (2000) found that some were reluctant to share their experiences due to the fear of ridicule, particularly when engaging with members of the scientific community. Additionally, in a study investigating descriptive qualities of ADC (called "Postdeath encounters"), one participant expressed, "I never told very many people about it because I didn't know how some people would take it" (Troyer 2014, p. 643). Such fears can result from a variety of factors such as the history and ongoing pathologization of ADC in the Western world, gender expectations (i.e., more reporting by women than by men potentially due to the greater acceptability of grief-related emotional expression among women in the West), adverse ADC experiences, and spiritual illiteracy on the part of mental health providers (For a recent review focused on the need for greater psychological attention to this topic, see Plante et al. 2023).

### 2.1. Pathologization

ADC has a long history of being viewed as a symptom of psychopathology (Exline 2021; Steffen and Coyle 2012; Wright 2008) and many individuals continue to cite the fear of pathologization for their nondisclosure (Pait and Exline 2023). In particular, some have expressed fear of being called crazy (Sanger 2009), being viewed as mentally unstable (Troyer 2014), or having the ADC interpreted as a hallucination (Pait and Exline 2023), delusion or other sign of psychosis (Parker 2005). Others may worry that ADC could actually be a sign of mental illness. Indeed, Sabucedo et al. (2020) conducted a study amongst mental health therapists to examine how they addressed ADC in treatment. Therapists in the study reported that they regularly saw clients who worried that they were going "mad" or would be called "mad" because of their ADCs.

Within the scientific literature, *after-death communication* is merely one term in a long list of scientific nomenclature used to describe the phenomenon. Other terminology includes *post-bereavement hallucination* (Rees 1971), *bereavement hallucination* (Kamp et al. 2019), *grief hallucinations* (Baethge 2002), and *post-bereavement hallucinatory experiences* (PBHE, Castelnovo et al. 2015). See Kamp et al. (2020) for a more detailed list of terms used to describe ADC. Terms like these may understandably stoke client hesitation while prompting mental health professionals to continue interpreting ADC as a symptom of aberrant thought processes. While Kamp et al. (2019) found that individuals who reported ADCs (called *bereavement hallucinations*) scored higher on indexes of psychological distress, leading the researchers to suggest that ADC may support the diagnosis of Persistent Complex Bereavement Disorder (PCBD), there is also ample evidence to suggest that ADCs also occur amongst healthy individuals who have not sought formal treatment nor felt the need to do so (Rees 1971). For example, Kelly (2002) conducted a study on police officers and

firefighters who experienced ADC from victims of fatal accidents. All study participants were found to be healthy and psychologically intact with no history of mental illness.

It is important to remember that such interpretations do not occur in a vacuum. It has been proposed that the pathologization of ADC is a consequence of a postmodern Western culture that has largely been dominated by the materialistic and mechanistic worldview of the sciences (Nowatzki and Kalischuk 2009; Plante et al. 2023). As a result of this, ADCs are often "explained away" as products of psychological or biological processes. Indeed, cross-cultural research on the topic has revealed that participants from cultures that accept and normalize ADC reported higher rates of encounters (Sabucedo et al. 2020). For instance, a study conducted in Japan found that 90% of widows reported feeling the presence of their deceased husbands (Yamamoto et al. 1969). The widows attributed this to the widespread tradition of ancestor worship and the belief that the departed remain present in everyday life (Yamamoto et al. 1969).

### 2.2. Gender Expectations

It may also be that gender expectations in the West influence the openness and willingness of men to report their ADCs. Several studies found that women were more likely to report ADCs than men (Daggett 2005; Elsaesser et al. 2020; Jahn and Spencer-Thomas 2014; Kalish and Reynolds 1973; MacDonald 1992). This has led some researchers to hypothesize that women may have more opportunities than men to share ADC experiences because the outward expression of grief-related emotions may be more socially acceptable for women (Elsaesser et al. 2020; Nowatzki and Kalischuk 2009). The socialization of men to stifle or ignore intuitive experiences may dissuade them from reporting ADCs when and if they have them (MacDonald 1992).

On the other hand, evidence has suggested that true differences between genders may occur between the types of ADCs perceived as opposed to the number of reported encounters. Kalish and Reynolds (1973) found that women reported more instances of intuitive contact or feeling the presence of the deceased whereas men reported more auditory or visual encounters. In addition, Klugman (2006) found that women reported more ADCs associated with hearing, feeling, seeing, and dreaming of the deceased whereas men reported more instances of smelling chemicals related to the deceased.

It is also possible that study demographics will shift, paralleling the continued evolution of gender roles occurring at the societal level. Troyer (2014), for example, conducted a more recent study investigating the ADCs of widowers and found the men to report a variety of encounters involving olfactory, visual, and auditory modes as well as experiences involving a sense of presence, countering earlier findings.

### 2.3. The Deeply Personal Nature of ADC

Despite the pathologization and potential gendering of ADC reporting, individuals may also choose to keep the encounters to themselves due to the deeply personal nature of the experiences (Pait and Exline 2023). Indeed, Barbato et al. (1999) and Nowatzki and Kalischuk (2009) found that study participants who felt comfortable sharing their ADCs only did so with trusted friends and family, as opposed to healthcare workers, demonstrating the intimacy and significance of the encounters. While some individuals may benefit from dissecting the details of their ADCs, others may worry that trying to explain or communicate such an ineffable experience in words will subtract from the mystery, beauty, and sacredness surrounding the phenomenon (Exline 2021; Longman et al. 1988). A related fear may be that mental health professionals will attempt to explain away the experience as just a dream or as a product of psychological mechanisms spurred by grief, minimizing it in the process (Exline 2021; Parker 2005). There is a tradition of reductionism in the field of clinical psychology; that is, the field has a history of explaining away religious and spiritual phenomena by presumably more basic psychological, social, and biological processes (Pargament 2002). The fears of disclosure among people who perceive ADCs may be a natural outgrowth of this bias in the field. In Parker's (2005) study on ADC and

adaptive outcomes of grief, one study participant expressed, "We [parents of deceased children] are not just going to tell anybody... It's hard to talk about... Cause I don't want anybody to sully it. I don't want anybody to mess with it..." (p. 276). Consequently, many view nondisclosure as a way to protect what they perceive to be deeply meaningful contact from the departed.

*2.4. The Dark Side of ADC*

The majority of people report ADCs to be comforting (McCormick and Tassell-Matamua 2016). In one study, 61.2% of participants, approximately 614 people, reported that having an ADC made it easier to accept the loss of their loved one (Elsaesser et al. 2020). However, some people find ADCs to be emotionally distressing and choose not to disclose them due to the negativity surrounding the events. Some felt their ADCs to be scary (Parker 2005), strange or shocking (Evrard et al. 2021), confusing (Jahn and Spencer-Thomas 2014), anxiety-provoking (Barbato et al. 1999), and, in some cases, abusive (Hayes and Leudar 2016). Lindström (1995) found that one-third of the study participants, 11 people, reported "extremely negative" ADC experiences soon after the deaths of their husbands, which led them to avoid objects or situations that could trigger subsequent encounters, including beds, bedrooms, their spouses' personal belongings, graves, or homes. Such negative experiences were associated with increased trait anxiety over time and difficulties in coping with bereavement (Lindström 1995). Elsewhere, 401 out of 819 (49.0%) individuals who lost loved ones to suicide reported feelings of overwhelming sadness and pain upon seeing or feeling the presence of the deceased so soon after their deaths (Jahn and Spencer-Thomas 2018). Additionally, Carlsson and Nilsson (2007) found that individuals who lost their spouses in palliative care felt stronger pangs of loneliness following their ADCs.

As a result of the fact that bereaved individuals are often emotionally vulnerable (LaGrand 1997, it is also quite understandable that perceived contact by the deceased may spark fear that one is being watched or haunted (Sabucedo et al. 2020). For example, one participant in Taylor's (2005) study on the counseling experiences of bereaved individuals was bothered by the idea that her deceased mother could see and hear everything about her personal life, including sex with her partner, ultimately leading to their separation. Another worried that two of her deceased partners were "comparing notes about their relationship to her" (Taylor 2005, p. 56). Others may feel that the experiences are upsetting or unwanted intrusions as they attempt to heal and move on with their lives (Sanger 2009).

Such experiences could also cause emotional distress if they are interpreted to mean that the departed is in pain or otherwise struggling in the afterlife. In a study investigating ADC in individuals bereaved by suicide, several participants indicated that the experience meant that their loved ones had not found peace or that they had regretted their final actions (Jahn and Spencer-Thomas 2018), understandably causing the mourners worry and sadness. It is also possible that ADC may not align with clients' spiritual or religious belief systems (Keen et al. 2013; Parker 2005) or that the phenomenon has catalyzed spiritual struggles (for a review, see Pargament and Exline 2022). For instance, many people experience blame or anger focused on God in response to the death of a loved one (Exline et al. 2011; Jahn and Spencer-Thomas 2018).

*2.5. Spiritual Illiteracy*

Finally, clients may not disclose their ADCs because they have met with adverse reactions upon sharing their encounters with mental health professionals in the past (see Roxburgh and Evenden 2016b for examples). Participants from Troyer's (2014) study reported negative counseling experiences largely characterized by clinician uninterest, deflection, pathologization of the topic, misconception of ADC as a grief response, or lack of inquiry into how ADC fits within personal belief systems. In his book, *Spiritually Integrated Psychotherapy*, Pargament (2007) outlined two types of spiritual disintegration on the part of mental health providers, including spiritual intolerance and spiritual illiteracy. In the case of ADC, the ongoing pathologization of the experience could be considered a

type of spiritual intolerance called rejectionism which may result when clinicians refuse to take spiritual matters seriously, shift focus to another topic, or negatively refer or respond to a client's religious belief (Pargament 2007).

The lack of clinical knowledge and skills to address ADC, as well as limited exposure to ADC-related topics in clinical training, demonstrate the ongoing problem of spiritual illiteracy in the field (Pargament 2007), an important issue to address given the common and widespread experience of ADC for bereaved individuals. The lack of knowledge and skills regarding ADC also points more broadly to the need for training designed to foster spiritual competency in mental health practice (Vieten et al. 2016). Religion and spirituality can be important aspects of cultural diversity and many major mental health disciplines and accrediting bodies, such as the American Psychological Association (2017), the American Counseling Association (see Ratts et al. 2016), the Association for Spiritual, Ethical, and Religious Values in Counseling (2023), the Council for the Accreditation of Counseling and Related Educational Programs (2023), the Commission on Accreditation for Marriage and Family Therapy Education (2021), the National Association of Social Workers (2017), and the Council on Social Work Education (2022), to name a few, have guidelines and/or ethical statements on the importance of training in these domains for mental health providers.

In fact, this problem has been noted by psychology trainees. In a qualitative study (*N* = 12) investigating the training needs of counseling and clinical psychology student clinicians regarding ADC and similar experiences, all of the study participants noted that they lacked the vocabulary to discuss such topics thoughtfully with clients (Roxburgh and Evenden 2016a). The student clinicians noted that gaining exposure to ADC and similar experiences through discussion groups, case studies, lists of relevant terms, and access to organizations and websites that disseminate information on the origins and prevalence of the events could help to normalize the experiences for them and make them feel better equipped to enter such conversations in therapeutic settings (Roxburgh and Evenden 2016a).

On the other hand, it may be that clients themselves struggle with finding appropriate or meaningful vocabulary to discuss their ADCs. ADC encounters can appear otherworldly or unusual, sometimes challenging the ways in which clients previously related to the world around them. There may have been sensations, experiences, or elements to the ADC that seem novel or unfamiliar. As a result of this, ADCs may seem ineffable and clients may not know how to begin sharing these experiences with others.

All of these factors (i.e., ongoing pathologization, Western scientific and medical paradigms, Western gender expectations, deeply personal encounters, and adverse experiences related to ADC) are likely to contribute to continued ADC nondisclosure (Pait and Exline 2023). Consequently, Houck (2005) has argued that ADC has fallen into the c by Doka (1999), in which individuals who believe they have been contacted by deceased loved ones are not allowed to share their stories of continued contact with the departed due to societal stigma. As a result, a driving force behind this review has not only been to investigate the roots of ADC nondisclosure but also to consolidate psychotherapeutic interviews and assessment strategies so that providers may develop knowledge and awareness of the phenomenon and the skills to effectively address it in therapy.

### 3. Interview and Assessment Strategies

Despite the obstacles behind nondisclosure, much evidence has demonstrated the therapeutic value of ADCs for bereaved individuals. For example, the majority of people who experience them report ADCs to be positive, comforting, reassuring, or helpful (Barbato et al. 1999; Daggett 2005; Drewry 2003; Elsaesser et al. 2020; Parker 2005; Streit-Horn 2011). In particular, words used by individuals to describe their ADCs have included *healing* (Nowatzki and Kalischuk 2009, p. 97), *forgiveness* (Drewry 2003, p. 81), and *connected* (McCormick and Tassell-Matamua 2016, p. 160). Bereaved individuals oftentimes interpret ADCs as opportunities to say final farewells or to resolve unfinished business with the de-

ceased (Parker 2005), to heal relational ruptures (Hayes and Leudar 2016), to reduce anxiety about or increase acceptance of death (LaGrand 1997; McCormick and Tassell-Matamua 2016; Parker 2005), or to form continuing bonds with their departed loved ones (Klugman 2006; Parker 2005). In a large, cross-cultural sample (*N* = 1004), Elsaesser et al. (2020) noted that the content of ADC messages was largely *reassuring*, *resolving*, *reaffirming*, and *releasing* (p. 15). Those who experience ambivalent, confusing, or distressing ADCs can also be assisted in therapy to help with their concerns (Sabucedo et al. 2020).

In spite of the disconnect between the potential therapeutic value of ADCs and the issue of client nondisclosure, how might mental health professionals best address the topic to help clients process and derive the most meaning from their encounters? Kamp et al. (2020) have offered a summary of general therapeutic strategies for clinicians working with ADCs, including an affirmative stance, psychoeducation, relationship processing, and addressing existential issues. Additionally, LaGrand (2005) has suggested four prerequisites for using ADC in the therapy space to help clients cope with the death of a loved one, including clinician familiarity with the topic, routine inquiry, techniques for validation, and knowledge of the client's belief system. The following sections aim to expand and deepen these therapeutic guidelines by offering specific suggestions and sample questions drawn from empirical studies and theoretical papers. A summary of the sample questions in this review is listed in Table A1, located in Appendix A. At this point, a hypothetical case example has been provided to set the stage for the second part of this review.

### 3.1. A Hypothetical Case Example

Scarlett is a 36-year-old seeking psychotherapy to help process the unexpected death of her mother, Lily, who died in a car accident six months ago. As an only child, Scarlett was very close with her mother up until a few years ago when she moved out of state with her husband. Lily, who had divorced Scarlett's father years earlier, did not adjust well to the change and routinely inquired if Scarlett would ever consider moving back to their hometown. Since her mother's death, Scarlett has felt extremely sad and guilty, blaming herself for leaving her mother on her own.

Midway through the third therapy session, Scarlett hints at an unusual experience she had the previous week that she would like to discuss but states that she feels hesitant about sharing it with a mental health professional. After gentle inquiry, Scarlett feels more comfortable discussing the details. She recounts that she had been drinking a glass of wine while sorting mail at the kitchen table one evening and glanced up to see Lily standing by the sink. Immediately shocked and frightened by the sight of her deceased mother, Scarlett froze in her chair. In the next moment, she felt as though Lily was communicating with her through her mind, as if telepathically, with all of the information coming in at once. She says that her mother wanted her to know that she was okay and at peace and that she had not felt any pain during the accident. But, Scarlett shares, "It was like I just knew these things automatically, rather than listening to my mom speak normally". She then felt a wave of immense and warm love before her mother disappeared completely. Although the encounter happened very quickly, Scarlett is certain that the experience was real, as opposed to a dream or a trick of her mind, but she has been scared to tell anyone for fear of being called crazy or overly dramatic. She is not sure what to make of the event but feels strongly that it holds great personal importance.

### 3.2. Clinician Self-Reflection

Clinicians may have varied responses upon hearing Scarlett's ADC report. Depending on their theoretical orientation, professional experiences, and personal beliefs, some elements may stand out more than others regarding Scarlett's personal history, presentation, and ADC recollection. One clinical inclination would be to label Scarlett's report of ADC as an isolated hallucination related to her alcohol consumption or a possible sign of psychosis. Another inclination may be to conceptualize the event as a visual representation of Scarlett's grief spurred by unexpected and complicated loss. A third inclination, prompted

by the clinician's personal religious or spiritual background, may lead them to interpret Scarlett's ADC as a real visitation from her mother's spirit or guardian angel. Each of these inclinations may impact the clinician's presence in the space, their relationship with Scarlett, and the goals for treatment.

We suggest that a first and key step for clinicians working with ADC may be to self-reflect on their own views and feelings about death, life after death, or other religious/spiritual beliefs that may influence their presence in the therapy space. Given that a large portion of mental health professionals report no religious affiliation (Koenig 2008) and a majority of psychologists report no belief in God (Shafranske and Cummings 2013), it makes sense that a clinician's reaction to ADC could influence how clients come to perceive the encounter. As LaGrand (1997) noted, even a therapist's silence and body language will make a statement.

Owing to this, clinicians should become aware of their natural inclinations toward ADC. Exline (2021) identified three lenses that clinicians may use to frame ADC: the *mental illness lens* (ADC as a symptom of disorder), the *psychological lens* (ADC as a result of normal psychological processes), and the *supernatural lens* (ADC as an actual communication from the deceased). Once aware of these inclinations, it may become easier for clinicians to hold client stories in an open, curious, and nonjudgmental manner (Kamp et al. 2020; LaGrand 2005). The following questions have been offered to help clinicians self-reflect on their natural inclinations toward ADC and other religious and spiritual experiences:

- Am I a religious or spiritual person? (Pargament 2007);
- What do I believe about death or what happens after death? (adapted from Pargament 2007);
- What is my honest and natural inclination toward reports of ADC? (adapted from Exline 2021);
- How might my personal beliefs, inclinations, and experiences impact my work with the client? (adapted from Exline 2021);
- Have I ever had an ADC or other spiritual or extraordinary experience? (adapted from Pargament 2007).

*3.3. Assess General Psychological Functioning*

Once clinicians have self-reflected on their natural inclinations toward ADC, it is important to assess the client's general psychological functioning, keeping in mind the reasons why the client has sought treatment (Kamp et al. 2020). Many studies have investigated ADC in the context of bereavement (Barbato et al. 1999; Carlsson and Nilsson 2007; Castelnovo et al. 2015; Daggett 2005; Datson and Marwit 1997; Elsaesser et al. 2020; Hayes and Leudar 2016; Houck 2005; Jahn and Spencer-Thomas 2018; Kamp et al. 2019; Keen et al. 2013; Lindström 1995; Longman et al. 1988; McCormick and Tassell-Matamua 2016; Parker 2005; Simon-Buller et al. 1989), but clients may also report ADC while seeking therapy for a host of other difficulties, such as in times of life crisis, major life transitions, or dealing with one's own death-related issues, for example. ADCs could be distressing for clients if they are related to underlying or preexisting mental health issues (Sabucedo et al. 2020). Clinicians should be sure to ask whether clients want to talk further about and explore their ADCs. In some instances, it might be of more interest to the clinician than the client, which may prompt the client's resistance. In other instances, even though they may have mentioned the ADC, the client may not feel ready to talk about it. In either case, the clinician is advised to respect the client's boundaries but state that they are open to discussing it if and whenever the client feels ready.

Additionally, there is always a chance that an ADC could be a symptom of psychopathology or the effect of a psychoactive substance (Exline 2021). As a result of this, it may be necessary to assess for signs of psychosis, substance use, dementia (Sanger 2009), complicated grief (Kamp et al. 2020), or trauma (Sabucedo et al. 2020). In Scarlett's case, she had been drinking a glass of wine when her ADC occurred. In a way that is neither accusatory nor invalidating, it would be reasonable to assess if she had ingested other

substances or medications that could have interacted with wine to affect her sight or to trigger a visual hallucination, such as a psychedelic substance. There may also be covert motivations for client disclosure, such as malingering, attention-seeking, or other secondary gains (Sanger 2009).

It Is also worth noting that other issues may develop in relation to ADC. For instance, some people who believe that their ADCs were real messages from the departed may seek continued communication with their loved ones via mediums or psychics and develop an addiction to calling psychics on expensive hotlines (Shepherd 2009). It is also plausible that some people might come to expect ADCs when their loved ones die and feel sad or disappointed if they do *not* experience one. Still, others could come to believe that they have special or magical abilities after having an ADC. Such beliefs, especially if coupled with symptoms such as maladaptive cognitive disorganization, anhedonia, and/or impulsive nonconformity, may suggest a leaning toward unhealthy schizotypy (Mason et al. 2005). Assessment questions for clinicians to keep in mind might include:

- Are there underlying mental health issues involved (adapted from Sabucedo et al. 2020)?;
- Is it possible that the client imagined or constructed the story (LaGrand 1997)?;
- Could substance use be involved (adapted from Sanger 2009)?;
- Might the client be malingering or using a report of ADC as an attention-seeking tactic (Sanger 2009)?;
- What other reasons might the client have for disclosing (adapted from Sanger 2009)?

These questions are intended to be internally guiding assessment questions for clinicians. Should clinicians feel the need to assess these domains overtly? Indeed, it is important to do so in a way that conveys curiosity and nonjudgmental acceptance and avoids expressing dis-belief or dismissal or pathologizing of the client's experience.

*3.4. Normalization and Validation*

If clinicians have assessed the client's general psychological functioning and ruled out ADC as a symptom of psychopathology, an important part of treatment may be clinicians' normalization of the event and validation that it could have occurred, particularly when considered veridical by the client (Exline 2021; LaGrand 1997; Nowatzki and Kalischuk 2009; Parker 2005; Sanger 2009). Indeed, it is a common experience for clients to disclose merely to seek validation that they have not gone crazy, so a natural next step may be to reassure the client that ADC is not typically a symptom of mental illness (Barbato et al. 1999; Exline 2021; Houck 2005; Keen et al. 2013; Sabucedo et al. 2020; Sanger 2009; Streit-Horn 2011; Taylor 2005). This occurs by allowing the story to unfold using active listening, empathy, and an affirmative stance (Barbato et al. 1999; Houck 2005; Keen et al. 2013; Kamp et al. 2020).

Clinicians may also relay that ADC is a common and well-documented phenomenon (Barbato et al. 1999; Sanger 2009). To this end, Streit-Horn (2011) provided an informative ADC Fact Sheet that may serve as a useful and psychoeducational tool. In a qualitative study on the counseling experiences of bereaved individuals who reported ADCs, Taylor (2005) found that participants felt therapy to be unsatisfactory when clinicians failed to understand, accept, or inquire about the nature of their ADCs or the relationships they had with the departed. One participant expressed that her counselor had made her feel "abnormal" because her experience did not align with the counselor's imposed stages-of-grief model (Taylor 2005, p. 58). Due to these issues, we suggest the following questions to help normalize and validate the encounters:

- What did you see/hear/feel/sense? (Taylor 2005);
- What was the departed person wearing/saying/doing? (LaGrand 1997);
- What words come to mind to describe the experience? (adapted from Streit-Horn 2011).

### 3.5. Assess the Client's Feelings

Given the spectrum of emotional reactivity to ADC, from uncertainty (Keen et al. 2013) or anxiety (Sabucedo et al. 2020) to comfort (McCormick and Tassell-Matamua 2016) or elation (LaGrand 2005), it is crucial to understand how the client feels about the encounter. ADC reports can be very clear and are often beneficial for experiencers (Parker 2005; Penberthy et al. 2021). For clients who feel excitement, curiosity, peace, or other similar positive emotions around ADC, encouraging an exploration into these emotions could help to reinforce the potentially healing aspects of the experience. However, a client's reactions to their ADC report may feel flooded with complicated or confusing emotions (see Hayes and Leudar 2016, for example). One way a therapist can offer support is by helping the client to identify and understand the differences between the actual ADC encounter and the client's reactions to it. Differentiating between the two can help the client to avoid bypassing uncomfortable or unfamiliar emotions that may prevent integration and meaning making. Example questions may be:

- How did you feel when it happened? (LaGrand 1997);
- How did you feel directly after? (LaGrand 1997);
- What emotions come up when discussing it now? (adapted from Greenberg 2015);
- How have your feelings about it changed? (Keen et al. 2013).

Based on client responses, clinicians will come to know whether the client has viewed the experience as positive, negative, neutral, or mixed with both positive and negative elements, as was the case with Scarlett. If positive, the ADC could serve as a healing mechanism to help the client process their loss and foster a healthy, continued relationship with the deceased (Klugman 2006; Parker 2005). In this case, psychotherapy that normalizes, accepts, supports, and explores the experience may be most useful (Sabucedo et al. 2020). If predominantly negative or distressing, the ADC may present an opening to explore the client's relationship with the deceased, unfinished business they had with the deceased, or pre-existing mental health issues (Houck 2005; Sabucedo et al. 2020). In this case, a grief stages model of psychotherapy may be suitable to help the client say goodbye and "let go of their loved one emotionally" (Sabucedo et al. 2020, p. 11). This approach may be appropriate if the predeath relationship was conflictual or experienced as problematic, in which case a continued bond with the deceased may be a source of strain (Root and Exline 2014).

If neutral or ambivalent, it may be that the client has viewed the ADC as nothing more than an anomalous experience that bears little to no weight on their grief process (Parker 2005). However, it may be necessary for clinicians to distinguish between true ambivalence and client apprehension because clients may never before have been granted permission to discuss the encounter. As Houck (2005) noted, it may take grieving clients time to adjust to a therapist's acceptance, empathy, and willingness to discuss a topic with such a stigmatized history.

### 3.6. Assess the Client's Explanation

Clients, like clinicians, may use a variety of conceptual lenses to interpret their ADC experiences (cf. Exline 2021) and responses to these questions may provide clues as to how clients have explained them. For example, does the client view it as real or as an illusion? Merely a dream or a tangible message from the departed? In a qualitative investigation of ADCs in a sample of widowers, Troyer (2014) learned that study participants explained their ADCs in three broad ways: believing it to be a trick of the mind, uncertain or possible contact from the deceased, or a real sign from the departed. One participant shared, "[I] said to myself, 'Your mind is playing tricks. She's not here. She's dead" (p. 642). Another concluded, "I cannot explain this... Because it was so real...I can't dismiss from my mind that it was real" (Troyer 2014, p. 642). Sometimes, the client may not know how to explain or interpret the event, only that it happened. In the case example, Scarlett felt certain that the experience of seeing her mother was real but had not yet come to a conclusion about

what exactly had occurred. Due to the spectrum of explanations, understanding the client's unique interpretation of their ADC can shape how the experience is addressed in treatment.

Some of these explanations may be rooted in the client's religious, spiritual, familial, or cultural traditions (see Exline and Pait 2022). For example, one widower in Troyer's (2014) study relayed, "...I do believe in God. I think it is something that is God's will. I think it is an indication that there is an afterlife of some sort" (p. 642). Keen et al. (2013) conducted a study on the presence of ADC in adults who had experienced bereavement in the last three years. One study participant shared, "...my family, like I said they all believe in this stuff, they call me special... in our world it's an everyday thing" (p. 352). As a result of this, it is important that clinicians pay careful attention to cultural considerations, client background, and personal history. According to the APA's multicultural guidelines, "Psychologists strive to conduct culturally appropriate and informed research, teaching, supervision, consultation, assessment, interpretation, diagnosis, dissemination, and evaluation of efficacy..." (American Psychological Association 2017, p. 5). Ramifications for explaining the event as a psychological response to grief, ignoring disclosure, or hastily interpreting it as a symptom of psychopathology could cause a relational rupture or severely disempower the client (Exline 2021; Sanger 2009; Taylor 2005). Sample questions to assess client explanations may be:

- What do you think about the encounter? (LaGrand 1997);
- What is your understanding of the event? (adapted from LaGrand 1997);
- How does the event fit within your culture? (adapted from Kamp et al. 2020).

### 3.7. Assess Client Spiritual Beliefs

Next, it may be beneficial to conduct a general assessment of the client's religious, spiritual, or afterlife beliefs. Pargament (2007) has constructed sample questions for an initial spiritual assessment to help clinicians gauge the salience and importance of addressing such matters in treatment. In addition, Christopherson and Beischel (2018) have constructed the Significance, Relationships, Resources, and Treatment (SRRT) afterlife assessment guide to help clinicians gain greater insight into client afterlife beliefs and spiritual experiences. Using open-ended questions in a conversational format, the guide explores how the four domains of significance, relationships, resources, and treatment may be related to afterlife beliefs in the context of religion and spirituality. In Scarlett's case, it could be helpful to gain greater insight into her afterlife beliefs by inquiring about her religious or cultural background and whether ADC is a common, expected, and accepted phenomenon or one that is stigmatized, feared, or ignored. Sample questions include:

- Do you see yourself as a religious or spiritual person? If so, in what way? (Pargament 2007);
- Has the experience affected you religiously or spiritually? If so, in what way? (Pargament 2007; Pargament and Exline 2022);
- Have you had other spiritual experiences that you would like to share with me? (Christopherson and Beischel 2018);
- Do you have any beliefs about the afterlife or about what happens after we die that you feel are important or that you would like to share with me? (Christopherson and Beischel 2018).

Exploring these beliefs will help clinicians to develop and use language that fits with, and is respectful of, the client's personal history and culture. General questions about religious, spiritual, and afterlife beliefs may also help clinicians to deduce the extent to which such beliefs may play a role in treatment, for better or for worse. As Pargament (2007) noted, sometimes spirituality is poorly integrated and may hinder, more than help, the therapeutic process. Clients may hold afterlife beliefs that are distressing, such as believing that a loved one is in Hell or inaccessible for an unknown reason. Such beliefs could contribute to divine spiritual struggles or struggles with doubt (Pargament and Exline 2022) as clients wrestle with these ideas.

*3.8. Meaning Making*

Once clinicians have assessed the client's feelings and explanations for ADC, as well as the client's spiritual beliefs, a crucial aspect in working with ADC is to help the client derive or cultivate meaning from the experience. This may concern existential or spiritual matters or the client's relationship with the departed. According to Park's (2010) Meaning Making Model, an individual's meaning system consists of two levels: global meaning and situational meaning. *Global meaning* refers to an individual's broad beliefs and feelings about how the world operates. This can include one's religious worldview, sense of purpose, and goals. *Situational meaning* refers to the meaning one ascribes to specific life stressors, such as illness, trauma, or death.

Park (2010) suggests that when an individual experiences an environmental stressor, they automatically attempt to appraise how and why the stressor occurred, how much control they have in the situation, and how threatened they feel. When an individual's situational meaning aligns with their broader, global meaning, they are more likely to successfully adjust to the stressor. On the other hand, if there is a discrepancy between the individual's situational meaning and global meaning, they may begin to experience distress. Meaning making is the process through which individuals attempt to reduce this discrepancy and come to accept the stressful event.

There are several recurrent themes in the literature regarding ADC and meaning making around the death of a loved one. First, ADC can help individuals to believe in the possibility of an afterlife, to confirm pre-existing beliefs of an afterlife, or to reinforce personal beliefs about death (Drewry 2003; Elsaesser et al. 2020; Jahn and Spencer-Thomas 2018; McCormick and Tassell-Matamua 2016; Nowatzki and Kalischuk 2009; Penberthy et al. 2021). In doing so, ADC could provide comfort and hope of a future reconnection with the deceased. Next, ADC could help individuals to maintain a form of continued attachment or relationship with the departed (Jahn and Spencer-Thomas 2018; McCormick and Tassell-Matamua 2016; Nowatzki and Kalischuk 2009). In this way, ADC may serve as a tool to help align an individual's situational meaning with their global meaning to reduce the distress caused by loss (Park and Benore 2004).

ADC also has the potential to prompt spiritual exploration, personal growth, or a renewed sense of life purpose (Drewry 2003; McCormick and Tassell-Matamua 2016). By ascribing a spiritual explanation for their loss, harnessing relational lessons, or rebuilding a sense of personal identity, individuals may be better equipped to accept and adjust to the event. In addition, Drewry (2003) noted that many individuals who had ADCs reported a reduced fear of death and a greater understanding of their place in the cosmos. Indeed, the phenomenon has been interpreted by many as a spiritual or sacred experience (Guggenheim and Guggenheim 1997; McCormick and Tassell-Matamua 2016; Pargament 2007) because it involves the connection to a transcendent realm beyond ordinary reality (Steffen and Coyle 2010). Penberthy et al. (2021) found that people who reported multiple ADCs endorsed an increase in religious beliefs after the events when compared to those who did not report multiple ADCs. Together, these findings demonstrate the potential impact of ADC on both situational and global meaning.

In Scarlett's case, helping her to make situational meaning from her ADC that aligns with her global meaning could help to reduce the pain she feels over her mother's death. First, the ADC might reassure Scarlett that her mother's soul is at peace, believing this could assuage Scarlett's guilt about moving away. Knowing that her mom was not in pain during the car accident could help Scarlett to release self-blame. Scarlett could also learn to reflect on the warm feeling of love that washed over her as a way to remain emotionally connected to her mom when she needs support. The experience could also impact Scarlett's perception of death or catalyze exploration of the nature of existence, perhaps giving Scarlett hope that she might reconnect with her mother again in the afterlife. If the goal of Scarlett's treatment is to adaptively process her mother's untimely death, her ADC may function to ease the pain of loss while simultaneously opening the door for personal growth. Sample questions to explore the meaning of ADC include:

- What is your deep intuitive feeling about the event? (LaGrand 1997);
- What do you think is the message (explicit or symbolic)? (LaGrand 1997);
- What does it mean for you? (LaGrand 1997);
- How does it help? (adapted from Taylor 2005);
- How does the event affect your relationship with the deceased? (adapted from Sanger 2009);
- Have there been any changes in your beliefs that you think are important to share with me? (Christopherson and Beischel 2018);
- How can we incorporate the wisdom you obtained from your spiritual experience into our work together? (Christopherson and Beischel 2018);
- Has your experience changed your sense of what is really important in life? If so, how? (adapted from Park 2013);
- Has your experience changed your understanding of your own purpose in life? If so, how? (adapted from Park 2013).

Based on the responses to these questions, clinicians may then help clients to deduce the best possible actions which may include devising a ritual to say good-bye to the departed (LaGrand 2005; Pargament 2007), seeking support from trusted others (Pargament 2007), or continuing to process the ADC at a later date (Taylor 2005). It may also be appropriate to offer clients religious or scientific resources should they desire further independent exploration (Christopherson and Beischel 2018). Other useful resources include support groups and organizations that help to explain ADC, such as the International Association for Near-Death Studies (see IANDS.org) and the Forever Family Foundation (FFF, see foreverfamilyfoundation.org). Support groups can provide a safe and moderated space where individuals can share their ADCs with others. This type of peer sharing can be helpful for processing, relating, validating, and problem solving to find language that assists in articulating the ineffable qualities of ADC. For example, the hearing voices movement (HVM, Higgs 2020) has a global mission to normalize experiences like hearing voices that can be related to ADC. The hearing voices network can provide resources, research, and support for individuals who report ADCs and similar experiences of a spiritual nature. Of note, IANDS, FFF, and HVM present ontological positions regarding ADC and similar phenomena that may not be appropriate for all clients. Thus, clinicians are encouraged to familiarize themselves with these organizations to assess whether they may be beneficial therapeutic resources.

## 4. Conclusions

The aim of this review has been to explore the roots of ADC nondisclosure and to present potential and specific interview and assessment techniques clinicians may employ to help bereaved clients process their encounters. ADC literature has revealed several sources of nondisclosure. Many individuals opt to keep their ADCs private, citing fears of pathologization. Men may not feel comfortable sharing due to Western gender expectations. Clients may perceive the ADC to be too personal to share with untrusted others who may minimize or explain away the experience. Still, others might have suffered a negative emotional response to the ADC, itself, or adverse reactions upon sharing the encounter with previous providers.

Due to these issues, it is important for mental health professionals to bolster their spiritual literacy by learning more about ADC and developing skills to address it in therapy. By self-reflecting on their personal views, clinicians can become aware of their biases and natural inclinations around ADC; these insights will hopefully help them to approach the subject in an open, curious, and nonjudgmental manner. By assessing a client's general psychological functioning, clinicians can examine (and perhaps rule out) the possibility that the ADC is a symptom of psychopathology or substance use. By normalizing and validating ADC as a common human experience, clinicians can create accepting and affirming environments in which clients feel comfortable disclosing these experiences. By assessing the client's feelings, explanations, and spiritual beliefs, clinicians can deduce

the best avenues for treatment. Finally, by guiding clients to cultivate meaning from their ADCs, clinicians can help clients to adaptively cope with the loss of a loved one, to process relational issues with the departed, to reinforce or further develop their spiritual belief systems, and to foster awareness and acceptance of death.

Avenues for future research may include the investigation of ways to develop spiritual literacy opportunities for mental health clinicians and healthcare providers alike so that they may appear as open and trusted others with whom clients feel comfortable disclosing ADC. In this vein, programs designed to cultivate spiritual competencies more generally among mental health practitioners have shown promising results (Pearce et al. 2020). Ultimately, ADCs can hold great personal and spiritual significance for bereaved individuals and clinicians have great potential to help them harness the possible and therapeutic benefits from these extraordinary encounters.

**Author Contributions:** Conceptualization, K.C.P. and J.J.E.; investigation, K.C.P.; resources, K.C.P.; writing—original draft preparation, K.C.P.; writing—review and editing, K.C.P., J.J.E., K.I.P. and P.Z. All authors have read and agreed to the published version of the manuscript.

**Funding:** This project was supported by a grant from the John Templeton Foundation, #59916.

**Institutional Review Board Statement:** The study was conducted in accordance with the Declaration of Helsinki, and approved by the Institutional Review Board (or Ethics Committee) of Case Western Reserve University (protocol code STUDY20210818 and date of approval 13 August 2021).

**Informed Consent Statement:** Not applicable.

**Data Availability Statement:** No new data were created or analyzed in this study. Data sharing is not applicable to this article.

**Conflicts of Interest:** The authors declare no conflict of interest. The funders had no role in the design of the study; in the collection, analyses, or interpretation of data; in the writing of the manuscript; or in the decision to publish the results.

## Appendix A

Please see Table A1 for a complete list of the interview questions listed throughout this manuscript.

**Table A1.** Psychotherapeutic Interview Questions to Assess Client Reports of After-death Communication (ADC).

| Psychotherapeutic Interview Questions to Assess Client Reports of After-Death Communication |
| --- |
| **Clinician Self-Reflection** |
| 1. Am I a religious or spiritual person? (Pargament 2007) |
| 2. What do I believe about death or what happens after death? (adapted from Pargament 2007) |
| 3. What is my honest inclination toward reports of ADC? (adapted from Exline 2021) |
| 4. How might my personal beliefs impact my work with the client? (adapted from Exline 2021) |
| 5. Have I ever had an ADC or other spiritual or extraordinary experience? (adapted from Pargament 2007) |
| **Assess General Psychological Functioning** |
| 1. Are there underlying mental health issues involved? (adapted from Sabucedo et al. 2020) |
| 2. Is it possible that the client imagined or constructed the story? (LaGrand 1997) |
| 3. Could substance use be involved? (adapted from Sanger 2009) |
| 4. Might the client be malingering or using a report of ADC as an attention-seeking tactic? (Sanger 2009) |
| 5. What other reasons might the client have for disclosing? (adapted from Sanger 2009) |

**Table A1.** *Cont.*

| Psychotherapeutic Interview Questions to Assess Client Reports of After-Death Communication |
| --- |
| **Normalization and Validation** |
| 1.   What did you see/hear/feel/sense? (Taylor 2005) |
| 2.   What was the departed person wearing/saying/doing? (LaGrand 1997) |
| 3.   What words come to mind to describe the experience? (adapted from Streit-Horn 2011) |
| **Assess Client Feelings** |
| 1.   How did you feel when it happened? (LaGrand 1997) |
| 2.   How did you feel directly after? (LaGrand 1997) |
| 3.   What emotions come up when discussing it now? (adapted from Greenberg 2015) |
| 4.   How have your feelings about it changed? (Keen et al. 2013) |
| **Assess Client Explanation** |
| 1.   What do you think about the encounter? (LaGrand 1997) |
| 2.   What is your understanding of the event? (adapted from LaGrand 1997) |
| 3.   How does the event fit within your culture? (adapted from LaGrand 1997) |
| **Assess Client Spiritual Beliefs** |
| 1.   Do you see yourself as a religious or spiritual person? (Pargament 2007) |
| 2.   Has the experience affected you religiously or spiritually? If so, in what way? (Pargament 2007; Pargament and Exline 2022) |
| 3.   Have you had other spiritual experiences that you would like to share with me? (Christopherson and Beischel 2018) |
| 4.   Do you have any beliefs about the afterlife or about what happens after we die that you feel are important or that you would like to share with me? (Christopherson and Beischel 2018) |
| **Meaning-Making** |
| 1.   What is your deep intuitive feeling about the event? (LaGrand 1997) |
| 2.   What do you think is the message (explicit or symbolic)? (LaGrand 1997) |
| 3.   What does it mean for you? (LaGrand 1997) |
| 4.   How does it help? (adapted from Taylor 2005) |
| 5.   How does the event affect your relationship with the deceased? (adapted from Sanger 2009) |
| 6.   Have there been any changes in your beliefs that you think are important to share with me? (Christopherson and Beischel 2018 ) |
| 7.   How can we incorporate the wisdom you obtained from your spiritual experience into our work together? (Christopherson and Beischel 2018) |
| 8.   Has your experience changed your sense of what is really important in life? If so, how? (adapted from Park 2013) |
| 9.   Has your experience changed your understanding of your own life purpose? If so, how? (adapted from Park 2013) |

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
