# Peer review of "After-Death Communication: Issues of Nondisclosure and Implications for Treatment"

_religions, doi:10.3390/rel14080985_

Round 1

Reviewer 1 Report

The manuscript is extremely well-organized, clear, concise, and thorough. It includes reference to much of the available literature on the topic of ADC. Those familiar with the literature know that although many of the studies cited are more than ten years, these are highly significant studies within the scientific research on ADC and there are a limited number of studies available within this specialized research area. This reviewer would encourage the author(s) to also review recent research finding by Elsaesser et al. (2020) (https://www.evelyn-elsaesser.com/wp-content/uploads/2020/02/Booklet_Web_English_Research.pdf and https://www.adcrp.org/_files/ugd/625c5c_71b6eb974fbf40a1b3a88c69ae4971bd.pdf), which may provide additional support from a current and very extensive questionnaire.

In the introduction (Section 1.), the author(s) succinctly summarize relevant research related to ADC and provide a clear explanation for supernatural attribution of these experiences, which strongly supports the need for mental health providers to have knowledge of ADC as well as guidance in therapeutically responding to client's disclosures of ADC. Likewise, the author(s) provide a compelling and well-supported justification for the need to develop an empirically-based framework for clinicians working with clients to integrate ADC experiences. 

The review of literature related to reluctance to disclose ADC is thorough and well-organized. Implicit within this section is important information for the reader related to what ADC experiencers need from mental health professionals if they are to feel safe to disclose their experiences and what they need in response to such disclosures.

It is notable that the manuscript discusses cultural factors (including Western influence and findings related to gender). This provides a valid basis for future inquiry and the importance of cultural competence of clinicians.

The discussion of distressing ADCs is helpful for clinicians to provide clear examples of what specific aspects related to ADC individuals have reported as distressing. Although it is mentioned that “many individuals have reported their ADCs to be comforting, others have found them to be emotionally distressing…” (Section 2.3, Lines 197-198), this reader would have liked to have some general idea of the number of participants in the cited studies that reported beneficial experiences related to ADC compared to the number of participants that reported distressing experiences related to ADC. In my clinical experience and review of the literature, ADCs tend to be experienced as beneficial far more often than distressing. This is addressed in Section 3, Lines 266-268, but may bear mentioning in Section 2.3 as well. The author(s) provide excellent context and specificity regarding what about the ADC experience created distress for participants. This provides excellent starting points for additional research.

The incorporation of spiritual illiteracy (Section 2.4) is an important component to address and it may be beneficial to relate this ethical mandates within various fields (psychology, counseling, social work, etc.) related to cultural competence. Addressing ADC within the disenfranchised grief paradigm provides a compelling rationale for the importance of developing a framework for addressing ADC effectively and ethically in a therapeutic context.

The inclusion of the Hypothetical Case Example (3.1) and Clinician Self-Reflection (3.2) is an exceptionally helpful practical example of a way in which a client may present in session. The case example could lead beautifully to a discussion in a classroom context of how the clinician might have intervened to facilitate the client feeling safer to disclose her ADC. The clinician self-reflection provides thoughtful and important questions for clinicians to consider when processing their own reactions and perceptions related to ADC. Reference to Exline (2021) is valuable at this point in linking the clinician self-reflection to the lenses clinicians use to frame ADC. With regard to the questions posed for clinician self-reflection (Lines 344-350), it may be helpful to extend the self-reflection a bit more into considering the impact on the client. For example, an additional question may be, “How might my beliefs, experiences, and inclinations impact my work with the client?”

Section 3.3 provides a necessary guide to helping clinicians effectively assess for psychological conditions that might need attention. The assessment questions (Lines 378-386) are extremely helpful considerations that could underlie disclosure. This reader clearly understands that these questions are meant to be internal guiding questions. However, it may be beneficial to include a statement related to the need to assess any of these areas in ways that are carefully worded to avoid conveying disbelief, pathologizing, or dismissing the client’s experience.

Section 3.4 provides valuable resource in the ADC Fact Sheet (Streit-Horn, 2011) and in the questions (Lines 410-412), as well as in reasserting the importance of normalization and validation of the client’s experience.

The statements on Lines 420-423 are exceptionally important for clinicians to understand and are very clearly stated with helpful examples. The inclusion of APA’s multicultural guidelines is an important inclusion. It may be beneficial to consider also including multicultural guidelines from the American Counseling Association (ACA) 2014 Code of Ethics, the Association for Spiritual Religious and Ethical Values in Counseling (ASERVIC) Competencies, and the National Association of Social Workers (NASW) Code of Ethics standards related to cultural competence.

The inclusion of section 3.7 is extremely helpful as assessment and intervention related to client’s religion and spirituality are underrepresented in clinical training programs.

As stated previously, the statement on Lines 521-523 is vitally important for clinicians to understand. The inclusion of Park’s (2010) Meaning Making Model with examples is extremely practical in providing ideas for how clinicians may help clients to integrate the ADC experience in a way that is congruent and meaningful for the client. As with previous sections, the inclusion of sample questions based in the research is extremely helpful.

Throughout the manuscript, the author(s) point the reader to additional resources, which is valuable to readers with little knowledge of the scope of the current research.

Mental health clinicians would greatly benefit from a framework such as the one provided in this manuscript to ensure the ethical and therapeutic efficacy of their responses to client disclosures of ADC.

This is one of the most well-organized, comprehensive, empirically-based, and practical manuscripts I have read related to the topic of clinically addressing ADC experiences. It provides invaluable information and guidance for mental health providers and would make an exceptional contribution to the field and as well as a training tool for clinical training programs.

This reader wishes to express gratitude to the author(s) for their excellent discussion of the topic coupled with practical suggestions for improving clinical care to clients who experience ADC. This is a valuable contribution to the field.

Reviewer 2 Report

This is a very nice and necessary article, useful for training and education about ADC.

I have some minor suggestions.

First, Kamp et al (2020, quoted) provided similar guidelines in high level psychiatric journal. Some of their ideas are included here, but their contribution is barely discussed. What are the differences here? Why are you not just endorsing their views?

A general view of exceptional/anomalous experiences is lacking here. ADCs are part of a larger spectrum of experiences for which issues like disclosure/non-disclosure and social stigma have already been discussed. I'm afraid a view only from ADC perspective may be too narrow. 

Schetsche, M. (2013). Pathologization as Strategy for Securing the Wirklichkeit The Example of Paranormal Experiences. DOI: 10.1007/978-3-531-18784-6_11

Palmer, G., & Braud, W. (2002). Exceptional human experiences, disclosure, and a more inclusive view of physical, psychological, and spiritual well-being. Journal of Transpersonal Psychology, 34(1), 29 – 59.

Regarding data about disclosure, you may rely on Valarino et al recent huge survey on ADC, disseminated through various publications, which provided data about disclosure, non-disclosure, and their reasons.

Especially, the article (where I confess being a co-author):

Evrard, R., Dollander, M., Elsaesser, E., Cooper, C., Lorimer, D., Roe, C. (2021). Exceptional necrophanic experiences and paradoxical mourning: Studies of the phenomenology and the aftereffects of frightening experiences of contact with the deceased. L’Evolution psychiatrique, 86(4), e1-e24. https://doi.org/10.1016/j.evopsy.2021.09.001

Adversive ADC were assessed in 108 participants.

Analysis of the 88 responses to open-ended questions Q140 and Q141 makes it possible to distinguish six categories, answering two questions: with whom did they share their experience? And how were they received?

The main category is: "With a few selected people" (79%); then "With everyone" (7%), and "With nobody" (3%). 

So selective disclosure is the main choice of disclosure, and this may be clearer in your paper. Further studies may clarify how "trusted ones" are selected and how health workers may fit with this description. 

The two extremes (with everyone / with nobody) can be questioned too, as they may reflect absence of opportunities, narcissistic issues, etc. 

This study confirms that healthcare professionals are rarely informed. 

Regarding the reception of their disclosure, this study provided some data too: Positive reception (by the trusted ones) is the main reception (51%), then mixed reception (32%), and negative reception (17%). Details are provided in the article.

The Interview and assessment strategies is a very nice part of the article.

Maybe an appendice can list and sequence all the questions? (a take-away note)

I was wondering about other issues that may appear: people believing they have special powers (like mediumship) after ADC; people urging to use facilitated or mediated ADC (with mediums or settings) after ADC, to pursue similar experiences; people becoming addicts to mediumship (some cases are published in the clinical literature). The after-ADC should be more investigated too.

p. 12: organizational suggestions of IANDS, FFF and HVM are not neutral enough to me. These are not public organizations with professionnals, but very activist organization with clear agenda (promoting continued bonds for instance, which don't fit for everybody). I will not refer my patients to them as recommended resources, but will offer also professional and scholar alternatives, while still stay open discussing their appraisal of these discourses. 

Recommendations for future research may include spontaneous selection criteria for "trusted ones" in such occurrences; and how to develop opportunities for healthcare workers to appear as skilled and "opened" to welcome such experiences. Besides, the whole procedure of having an active listening of such experiences instead of Practice-as-usual can be tested for its effectiveness (both in providing more testimonies and more help).

p. 3: small mistake

police offers à officers

Reviewer 3 Report

Dear Authors,

I thoroughly enjoyed reading your paper. I have researched these experiences myself for the past 15 years and have published a number of papers and book chapters on the topic. I thought you not only demonstrated a very good grasp of the literature (with some exceptions, see below), your review summarised much of this literature really well, with a useful focus on reasons for non-disclosure and clinical implications; it was nice to see some attention to spiritual meaning-making, especially as relevant for a publication in Religions. Perhaps this aspect could have been extended but I would not state this as a request for change, merely as a wish.

I would also like to congratulate you on the excellent writing. I cannot even count the number of papers I have been sent for peer review on this topic that are poorly written, with a poor grasp of the literature etc. Your paper was not only a breath of fresh air but a model of how it should be done. There is perhaps one small section that stands out a bit from a writing perspective as not quite aligned in style with the rest - this is when you suddenly address clinicians directly, speaking to them with 'you' and making assumptions about how they might react when being faced with such experiences. I would recommend changing this to be brought in line with the rest of the otherwise very academic and professional tone of the review. Being not only a researcher but also a clinician myself, I found it just a little bit patronising.

Re literature that was missing, I don't think you cited Castelnovo et al.'s review, which was the most authoritative review until the more recent one by Kamp et al. I think this is an omission and I would suggest including it. Furthermore, I did not see any reference to Woollacott et al., Elsaesser et al., (Chris Roe, Callum Cooper etc.) who have recently published a number of papers all about ADCs, and this is an even more glaring omission in my view. 

Thus, I have 4 requests for amendments:

1. To change the style of the section addressing clinicians

2. To include/reference Castelnovo et al.

3. To refer to the recent works by Woollacott, Elsaesser and the rest of their team.

4. The chapter you cite by Steffen & Coyle from 2012 in the text does not appear in the references at the end.

Reviewer 4 Report

This is an interesting paper that makes a valuable contribution to the clinical literature on so-called after-death communications (ADCs). A persuasive case is made for the ontologically neutral stance that is adopted, so that the focus can be on the experience itself, its interpretation and impacts. The authors outline some of the standard concerns experiencers may have about disclosing an ADC, which might dissuade them from sharing their experiences with others, including the fear of pathologisation, fear of ridicule, or concern that the phenomenon was too personal or potentially upsetting to disclose to others. The analysis of a hypothetic case study is a useful way of rehearsing the issues involved in supporting a client who present such an experience. Generally, the treatment of this controversial phenomenon is clear and balanced, though in places there is a tendency to underplay the potential beneficial effects, as outlined below. Space constraints may preclude a comprehensive review of relevant literature; nevertheless, there are some publications that in my view are directly relevant to the case being made (details given below), and the paper would benefit from their inclusion.

For example, in the section on spiritual illiteracy, passing mention is made of therapists’ limited exposure to this kind of phenomenon as part of their clinical training, leaving them to feel out of their depth, so that attempting to bracket the ADC disclosure is more of a defence mechanism than their ‘buying into’ a particular materialist worldview. See Roxburgh & Evenden (2016a).

Similarly, Kamp et al.’s (2020) summary of therapeutic strategies that could be applied to supporting people who disclose ADCs is a promising framework within which to work (p 6). Of course, not much of this is new, and in fact has been applied generally to ‘anomalous experiences’ that seem to contradict basic assumptions of the prevailing worldview and so trigger the same kinds of concern in clients. A useful resource is Kramer, Bauer, & Hövelmann’s (2012) edited volume, Perspectives of clinical parapsychology: An introductory reader Stichting Het Johan Borgman Fonds. A pdf of the book is available at https://www.parapsych.org/blogs/wkramer/entry/80/2015/10/free_pdf_of_book_on_clinical.aspx

Given the clinical focus of this narrative review, it is surprising that the authors don’t mention the following:

Evenden, RE, Cooper, CE, & Mitchell, G. A counselling approach to mediumship: Adaptive outcomes of grief following an exceptional experience. Jour Exceptional Experiences and Psychology 2013, 2(1), 14-23.

Roxburgh, EC, & Evenden, RE. (2016a) ‘Most people think you’re a fruit loop: Clients’ experiences of seeking support for anomalous experiences’. Counselling and Psychotherapy Research, 16(3), 211–221. https://doi.org/10.1002/capr.12077

Roxburgh, EC, & Evenden, RE. (2016b) ‘It’s about having exposure to this’: investigating the training needs of therapists in relation to the issue of anomalous experiences, British Journal of Guidance & Counselling, 44 (5), pp. 540-549. ISSN 0306-9885

Taylor, SF. Between the idea and the reality: A study of the counselling experiences of bereaved people who sense the presence of the deceased. Counselling and Psychotherapy Research 2005, 5, 53–61.

The section on the dark side of ADCs (p 5), while necessary for balance, is overstated, particularly when the authors go on to say (p 6) “the majority of people who experience them report ADCs to be positive, comforting, reassuring, or helpful”. Lindstrom (1995) is hardly representative of the extant literature and some of the other sources in this section are unable to distinguish between the effects of ADCs and effects of losing a loved one per se. Elsaesser et al. is more typical in finding that while ADCs might have been frightening y virtue of being unexpected and not necessarily fitting in with the experient’s worldview, they are nevertheless subsequently highly valued and meaningful.

p. 5: The authors state “negative counseling experiences largely characterized by clinician disinterest, deflection, or pathologization of the topic” -- although it is becoming more common to use disinterest as a synonym for uninterest, I think this should be avoided in academic writing.

The section Assess Client Feelings (p 9) seems unnecessarily negatively skewed given the findings that have already been reviewed, which indicate that ADCs can be hugely beneficial if experiencers are given the opportunity to reflect and process. Of course, it is proper to consider the possibility of pathology (albeit within the context that the literature does not give any support for the idea that they are symptomatic, and the experiences are so widespread it does not seem meaningful to consider them to be abnormal), but only two sentences are devoted to the possibility that ADCs can be a positive opportunity for healing before we quickly return to their being predominantly negative or distressing. The literature strongly suggests that people who have ADCs experience better outcomes to their bereavement process. For examples, see the following:

Hayes J, Leudar I. Experiences of continued presence: on the practical consequences of ‘hallucinations’ in bereavement. Psychol Psychother Theory Rese Pract 2016; 89: 194–210.

Penberthy, J.K., Pehlivanova, M., Kalelioglu, T., Roe, C.A., Cooper, C.E., Lorimer, D., & Elsaesser, E.  (2021). Factors moderating the impact of after death communications on beliefs and Spirituality. Omega: Journal of Death and Dying, 1-18. DOI: 10.1177/00302228211029160

Steffen E, Coyle A. ‘Sense of presence’ experiences in bereavement and their relationship to mental health: a critical examination of a continuing controversy. In Mental Health and Anomalous experience (ed. Murray C): 33–56. Nova Science Publishers, 2012.
